# The Electrochemical Behavior of Unmodified and Pd-NPs Modified AB_5_ Hydrogen Storage Alloy in Selected Protic and Aprotic Ionic Liquids (ILs): Towards ILs-Based Electrolytes for Ni-MH Batteries

**DOI:** 10.3390/molecules28020856

**Published:** 2023-01-14

**Authors:** Katarzyna Hubkowska, Małgorzata Pająk, Michał Soszko, Andrzej Czerwiński

**Affiliations:** Faculty of Chemistry, University of Warsaw, Pasteura 1, 02-093 Warsaw, Poland

**Keywords:** AB_5_ alloy, ionic liquid, palladium nanoparticles, hydrogen sorption, 1-ethyl-3-methylimidazolium methanesulfonate, diethylmethylammonium triflate

## Abstract

The objective of this work was to study the electrochemical behavior of AB_5_ alloy and its composite with Pd nanoparticles in selected ionic liquids. The protic ionic liquid (diethylmethylammonium triflate) and the mixture of aprotic ionic liquid (1-ethyl-3-methylimidazolium methanesulfonate) with parent superacid were used as electrolytes in the process of hydrogen electrosorption in AB_5_ alloy electrodes. The impact of the surface modification of AB_5_ electrode with Pd nanoparticles has been checked. The studies revealed that the highest hydrogen absorption capacity can be obtained in Pd-NPs-AB_5_ electrode in DEMA-TFO. It was found that the surface modification with Pd-NPs facilitates the activation of the electrode and results in stabilization of the plateau potential of discharging. The studies show that more effort should be put into the synthesis of less corrosive tailored ionic liquids suitable to be used as electrolytes in hydride batteries.

## 1. Introduction

The increase in world energy demand and consumption of fossil fuels have a negative impact on environment and energy resources. Therefore, there is a need to develop new energy-saving technologies in an environmentally responsible way. The development of next-generation rechargeable batteries seems to be crucial for replacing the energy strategy based on fossil fuel [1,2]. So far, several types of rechargeable, secondary batteries, such as nickel-metal hydride (Ni-MH), lithium-ion (LIB), and lead-acid batteries have been practically applied [3,4]. Currently, the market is dominated by LIBs as they have relatively high energy density but cannot be used without electronic control for safety reasons, especially in electronic vehicles and military electronic devices [3,5]. Ni-MH batteries are more environmentally friendly than LIBs, have high power capacity, and tolerance to overcharge/discharge but they also have some disadvantages. The electrochemical behavior of Ni-MH batteries strongly depends on the electrode material properties. The AB_5_-type alloy is the most popular and marketable anode material for Ni-MH due to its high hydrogen capacity at room temperature. However, the greatest problem of this type of alloy is high corrosion rate in alkaline solution, pulverization caused by lattice expansion/contraction during hydrogenation, and poisoning of the alloy surface which influences the life cycle and the capacity of Ni-MH battery [6]. These negative effects can be minimized by appropriate electrode material and electrolyte modifications.

The typical method of the improvement of the poisoning tolerance and to enhance hydrogen absorption is related to surface modification by introduction of platinum group metals, especially Pd. The catalytic effect of Pd on hydrogen absorption has been extensively studied over the past decades [7,8,9,10]. Palladium catalyzes hydrogen evolution reaction, promotes activation, and increases poisoning tolerance. Moreover, the palladium layer is permeable to hydrogen but impermeable to larger molecules. Our previous study on surface decoration of AB_5_-type alloy with small amount of Pd nanoparticles (ca. 3.5 wt% Pd-NPs) [11] showed better electrochemical properties towards hydrogen sorption of such material in comparison with pristine AB_5_ alloy due to enhanced kinetics. 

The second method to improve the electrode performance involves electrolyte modifications. Ruiz et al. [12] studied the effect of the electrolyte concentration on the electrochemical properties of AB_5_-type alloy and found that electrolyte concentration affects the activation and discharge capacity. Yan et al. [13] investigated the effect of salt additives to KOH electrolyte and revealed that the corrosion efficiently decreased. Moreover, polymer and gel electrolytes were tested as promising candidates for application in Ni-MH batteries that can enhance the life cycle performance by preventing the solution leakage problem and inhibiting the cell deflection [5,14,15]. Recently, ionic liquids (ILs) have been intensively studied as possible alternatives to conventional electrolytes due to their unique physicochemical properties involving high chemical and thermal stability, low vapor pressure, non-flammability, and wide electrochemical window [16,17]. In the last years, a large number of studies have been dedicated to ILs as electrolytes in LIBs [18,19,20], but there were only a few reports on ILs-based electrolytes for Ni-MH batteries. Young et al. [21] and Meng et al. [22] applied aprotic IL with non-aqueous acid to supply the protons to carry the charge; however, they tested mixtures of various aprotic ILs (AILs, differing in cation and anion) with a weak organic acid—acetic acid (HAc). Our previous studies on hydrogen sorption measurements with the use of palladium limited volume electrode (Pd-LVE) as a model system for Ni-MH [23,24,25,26] revealed that protic ILs (PILs) can be successfully used as electrolytes in such experiments. Therefore, the aim of this study was to use neat PIL and AIL with parent superacid additive as electrolytes in electrochemical behavior investigation of the unmodified and Pd-NPs decorated AB_5_-type alloy. Based on results obtained by us earlier [23,24,25,26], we chose PIL with the best properties for hydrogen sorption measurements—diethylmethylammonium triflate (DEMA-TFO) and AIL 1-ethyl-3-methylimidazolium methanesulfonate (EMIm-MS) with methanesulfonic acid (HMS) additive. The structures of the electrolyte components used in this study are presented in Figure 1. 

## 2. Results

### 2.1. Physicochemical Properties of DEMA-TFO and 1 M HMS/EMIm-MS

The physicochemical properties such as dynamic viscosity, density, ionic conductivity, and water content of investigated ILs are shown in Table 1. DEMA-TFO and 1 M HMS/EMIm-MS differ strongly in dynamic viscosity which can affect the results of the electrochemical measurements. The viscosity of 1 M HMS/EMIm-MS is ca. 3 times higher than the viscosity of DEMA-TFO. This can result in hindered proton transport and thereby in relatively high irreversibility of the hydrogen electrosorption signals, which has been observed in the process of hydrogen electrosorption in Pd-LVE electrode in IL media [23,26]. Ionic conductivity of 1 M HMS/EMIm-MS is lower than DEMA-TFO, which is related to high dynamic viscosity of the former. The values of the cathodic and anodic limit indicate that the ILs-based electrolytes are stable in the applied potential range. Moreover, Table 1 demonstrates that drying of the ILs under the molecular sieves enable to efficiently reduce the water content to a value not exceeding 700 ppm.

### 2.2. Electrochemical Behavior of AB_5_/AB_5_-Pd-NPs in DEMA-TFO and 1 M HMS/EMIm-MS 

Figure 2 presents cyclic voltammetry behavior of AB_5_ and Pd-NPs modified AB_5_ electrode in two IL-based electrolytes, i.e., protic ionic liquid (DEMA-TFO) without any additives and aprotic ionic liquid (EMIm-MS) with the addition of the precursor acid. It can be noticed that the type of electrolyte as well as the modification of the electrode surface influence on the cyclic voltammetry (CV) behavior of AB_5_ alloy. For pure AB_5_ in DEMA-TFO (Figure 2a) before the activation process, only the signal of hydrogen evolution is visible.

However, after the process of continuous cycling in the IL electrolyte, more signals are developed: a signal of hydrogen oxidation from AB_5_ in the potential range ca. 0.2–1.2 V and characteristic, reversible signals from gaseous hydrogen reduction and oxidation (−0.8–0 V). The latter was also noticed when hydrogen was electrosorbed in Pd thin film electrode from DEMA-TFO [23,26]. The signals from gaseous hydrogen are present only for protic ionic liquids and they are better visible for electrodes after long cycling in the hydrogen region [26]. During hydrogen absorption, the AB_5_ particles break into smaller ones, which causes the increase of surface area and results in more significant generation of gaseous hydrogen. Therefore, these signals are greater for AB_5_ modified with Pd-NPs, where the surface area is larger than in unmodified AB_5_. Moreover, ILs are characterized with relatively high dynamic viscosity which hinders the removal of gaseous hydrogen from electrode surface. The CV behavior of AB_5_ in 1 M HMS/EMIm-MS (Figure 2b) differs from the one registered in DEMA-TFO. One can notice that in 1 M HMS/EMIm-MS, already in the first CV cycle, signals from hydrogen absorption and hydrogen oxidation are developed. In addition, in 1 M HMS/EMIm-MS, the signals from hydrogen absorption and hydrogen evolution reaction are better separated than in DEMA-TFO. After the activation process, the signals from hydrogen absorption and desorption are higher, indicating that the process of hydrogen absorption declines in the depth of the active material. The CV behavior of AB_5_ and Pd-NPs-modified AB_5_ electrode significantly differ from the one obtained in standard alkaline electrolyte (6 M KOH; see [11]). In the alkaline electrolyte, one can distinguish two main, well-developed signals originating from Pd-NPs containing phase: the reduction at −0.02 V and the oxidation at 0.11 V. For pure AB_5_ alloy in 6 M KOH, the hydrogen absorption occurs at ca. 0.02 V, whereas the hydrogen is oxidized at ca. 0.25 V. The modification of AB_5_ with Pd-NPs in alkaline electrolyte causes the activation procedure of the AB_5_-Pd-NPs electrode to not be required, since the CV behavior is the same for the activated and not-activated electrode. Comparing the results for aqueous alkaline media and ILs one can notice that in 6 M KOH, more reversible, better developed signals from Pd-NPs containing phase can be distinguished and the obtained currents are higher than in ILs media. Moreover, for the electrodes before and after the activation, the reduction signal from the Pd-NPs containing phase is not visible. The oxidation signal from this phase is visible before the activation process; however, after the activation, it overlaps with the oxidation signal from the AB_5_-containing phase. It should be underlined that the hydrogen oxidation signal from Pd-NPs-containing phase is present at similar potential values in ILs and aqueous alkaline media, whereas there are significant differences in the case of the AB_5_-containing phase (lower potential for absorption and higher potential for desorption in ILs compared to 6 M KOH).

However, it is remarkable that after the activation of AB_5_ in 1 M HMS/EMIm-MS, two oxidation signals can be distinguished. The presence of the two hydrogen oxidation signals can be assigned to the segregation of the hydrogen absorbing phases after the hydrogen absorption in 1 M HMS/EMIm-MS. EDS (Table 2) and EDS mapping results (Figure 3, Appendix A) seem to confirm this prediction since after hydrogen absorption in AB_5_ in 1 M HMS/EMIm-MS more Ce and Nb occurs near the electrode surface. However, XPS results show that Ce and Nb are not present at the electrode surface, which indicates that the segregation of the metals occurs in the alloy grain below several atomic layers. The hydrogen oxidation signal originating in more negative potential can be assigned to the hydrogen absorbing phase containing more Ce and Nb. It is worth noting that the metal segregation process in AB_5_ particles starts just after the synthesis, which is a well-known phenomenon for AB-type alloys obtained by ball-milling. La, Ni, and Co tend to segregate near the electrode surface, whereas Ce and Al are present only in the depth of the grains. The obtained results denote that hydrogen absorption in 1 M HMS/EMIm-MS influences the segregation of metals in AB_5_, causing higher concentration of Ce and Nb near the surface. 

The CV behavior of Pd-NPs modified AB_5_ electrodes (Figure 2c,d) differs significantly from the one of the unmodified AB_5_. The differences are more pronounced for the first CV cycle. In both ILs electrolytes, hydrogen oxidation (hydrogen desorption) signals from Pd-NPs are visible (ca. 0.1 V). After the activation process in DEMA-TFO (Figure 2c), the signal from hydrogen reduction (hydrogen absorption) in Pd-NPs can also be distinguished (ca. 0 V). The most prominent effect is for the electrodes after the activation process since the max. values of the oxidation currents increase ca. two times for Pd-NPs modified electrodes compared to the unmodified ones. The observed behavior results in the significant differences of hydrogen sorption capacities registered for unmodified and Pd-NPs modified AB_5_ electrode in PIL and AIL/acid. Figure 4 shows chronopotentiometric charging and discharging of the electrodes in ILs media. Two types of electrodes have been charged to the value of max. theoretical capacity of AB_5_ (i.e., 290 mAh g^−1^) [11]. It can be noticed that in 1 M HMS/EMIm-MS (Figure 4b), the presence of Pd-NPs on the electrode surface influences on the plateau potential of charging. In 1 M HMS/EMIm-MS, the AB_5_-Pd-NPs electrode can be charged in the significantly higher potential value than unmodified AB_5_. Contrastingly, in DEMA-TFO (Figure 4a), the modification of the electrode does not influence the charging potential. The modification of the electrodes with Pd-NPs influences more the discharging process. These differences in the plateau of charging can be assigned to the additional processes occurring at the electrodes. Comparison with the CV results (Figure 2) allows us to state that the process of the electrochemical activation of the electrode influence on the decrease of the hydrogen electrosorption potential is only in the case of alloy in 1 M HMS/EMIm-MS (both for Pd-NPs modified and unmodified electrode). The electrochemical activation in DEMA-TFO does not influence the charging potential since at the higher potential values, there is generation of gaseous hydrogen hindering the hydrogen sorption.

In the case of both electrolytes (Figure 4c,d), the presence of Pd-NPs on the AB_5_ surface results in the stabilization of the plateau potential of discharging and increase of the hydrogen capacity. However, one should notice that when the AB_5_-Pd-NPs electrode is discharged in DEMA-TFO (Figure 4c) to the potential of 1.2 V, two plateau can be distinguished on the discharging curve. The comparison of chronopotentiometric discharging curve with the cyclic voltammetry reveals that the presence of two plateau originates from significant difference in discharging potential of hydrogen from the Pd-NPs-modified AB_5_ phase and pure AB_5_ phase. It was not observed in the case of 1 M HMS/EMIm-MS where both oxidation signals are less separated.

### 2.3. Hydrogen Absorption Capacities

In Table 3, hydrogen absorption capacities of electrodes calculated from the discharging curves are shown. In the case of DEMA-TFO, the contribution from the gaseous hydrogen has been subtracted from the total charge.

Interestingly, similar values of the capacities have been obtained for AB_5_ in both electrolytes. In comparison with standard electrolyte (6 M KOH) which is based on water, IL electrolytes demonstrate poor wettability of alloy grains. Thus, only the part of the AB_5_ grains is active in the process of hydrogen electrosorption, which results in reduced hydrogen absorption capacity. The modification of AB_5_ with Pd-NPs increases the hydrogen absorption capacity few times compared to the unmodified electrode. The presence of Pd-NPs on the surface of AB_5_ particles significantly facilitates the penetration of hydrogen into the depth of the AB_5_ grains and results in higher hydrogen capacities, constituting ca. 69% of the nominal capacity of AB_5_ alloy) and ca. 85% of the experimental capacity measured in similar conditions in 6 M KOH (i.a. the electrode without binding additives). Every AB_5_ particle is surface-covered with thin oxide film, which hinders the process of hydrogen electrosorption. The contact with corrosive electrolyte facilitates the hydrogen sorption since the surface oxides are removed. Moreover, the process of corrosion occurs only at the surface of the alloy—it does not progress deep into the grain. This result can be compared with the one obtained by Meng et al. [22] for whole cell based on the AB_5_ anode and sintered Ni(OH)_2_ cathode in the mixture of 2 M HAc and EMIm-Ac. The maximum capacity of the cell equals to ca. 162 mAh g^−1^ (at charge current 4 mA g^−1^). It should be emphasized that reported maximum capacity was obtained with the use of unmodified AB_5_ and measured for the cell.

### 2.4. Physicochemical Analysis of AB_5_ Electrodes

The Pd-NPs modified and unmodified AB_5_ alloys have been subjected to physicochemical analysis such as an ex situ X-ray photoelectron spectroscopy (XPS) and scanning electron microscopy (SEM) with energy dispersive spectroscopy (EDS) before the electrochemical treatment and after hydrogen electrosorption performed in DEMA-TFO and 1 MHMS/EMIm-MS. 

EDS results (Table 2) show that there is no significant change in the electrode composition before and after Pd-NPs modification and after the electrochemical treatment in PIL/AIL electrolyte. However, EDS gives information from ca. 1 µm of electrode depth. Thus, the electrodes have been subjected to XPS analysis (Table 4), giving information from few surface layers. It is worth noting that the presence of Pd-NPs and high amount of carbon and oxygen on the electrode surface can interfere with the results. However, comparison of EDS and XPS results confirms general statement, concerning segregation of the metals in pristine AB_5_ grain: surface enrichment in La, Co, and Mn and the absence of Al and Ce near the surface. Meli et al. [27] identified contact with oxygen as a driving force behind the segregation of metals in pristine AB_5_ alloy. The analysis of the SEM images (Figure 5) reveals that the morphology of the electrode changes after electrochemical treatment in PIL/AIL media. The most significant changes can be noted for AB_5_ electrode after hydrogen electrosorption in DEMA-TFO (Figure 5e).

On the surface of the AB_5_ alloy, apart from surface cracks originating from hydrogen sorption, there are many pits. However, it has not been observed for the Pd-NPs modified samples. Thus, it reveals that the AB_5_ alloy modification with Pd-NPs limits the surface corrosion originating from the contact with IL electrolyte. In relation to SEM images, DEMA-TFO seems to be more aggressive in contact with electrodes than 1 M HMS/EMIm-MS. In the case of 1 M HMS/EMIm-MS binary system, [MS(HMS)]^-^ clusters are formed and HMS hydrogen atom is H-bounded within such cluster [28]. This process lowers the acidity of the HMS proton and therefore mixture of AIL with its parent superacid is less aggressive to the electrode surface. Moreover, the H-bounded HMS proton is less available in the hydrogen electrosorption process, which in turn leads to lower capacity values obtained in 1 M HMS/EMIm-MS electrolyte. In Table 4, surface atomic concentrations of different elements for the AB_5_, AB_5_-Pd-NPs pristine and after treatment in both ILs are shown. Analysis of the XPS results do not confirm unequivocally trends in metal dissolution. Moreover, one should note that surface composition of the electrodes after electrochemical treatment can be affected by two processes: surface dissolution and hydrogen electrosorption-induced segregation of metals. Table 4 shows changes in surface composition mostly in the case of La and Co. Interestingly, after hydrogen electrosorption in both IL electrolytes, relatively high Al amounts are present on the surface. It indicates that the process of hydrogen electrosorption in ILs influences the segregation of Al atoms towards the alloy surface. The opposite effect was observed after hydrogen electrosorption in AB_5_ in 6 M KOH—there were no Al on the surface, whereas the presence of Ce has been noted. 

## 3. Conclusions

It has been shown that IL media can be effectively used as electrolytes and proton donors in the experiment of hydrogen electrosorption in AB_5_-based electrodes. The research revealed that the surface modification of AB_5_ with Pd nanoparticles influences significantly the max. capacities of AB_5_ electrodes both in PIL and AIL. The highest max. hydrogen capacity was obtained for Pd-NPs modified AB_5_ electrode in DEMA-TFO. It constitutes ca. 69% of the theoretical hydrogen capacity of AB_5_ electrode. Moreover, the surface modification with Pd-NPs facilitates the process of the electrode activation and results in the stabilization of the plateau potential of the electrode discharging. The results of the described preliminary research indicate that the greatest effort should be made to synthesize tailored ionic liquids suitable for the efficient performance of the electrode material. 

## 4. Materials and Methods

### 4.1. Materials

Diethylmethylammonium triflate, DEMA-TFO (>98%) and 1-ethyl-3-methylimidazolium methanesulfonate, EMIm-MS (99%) were received from IoLiTec (Heilbronn, Germany). Methanesulfonic acid, HMS (≥99%) was purchased from Sigma-Aldrich (St. Louis, USA). Gold wire (99.99%), gold plate (99.99%), and silver wire (99.9%) were obtained from Mint-Metals (Radzymin, Poland). Gold mesh (99.9%, wire diameter 0.06 mm, open area 65%) was from Goodfellow (Huntingdon, England). Before the electrochemical measurements, DEMA-TFO and the mixture of EMIm-MS with HMS were dried for a few days under the molecular sieves (type 4A, Chempur, Piekary Śląskie, Poland) until the water content (KF titration) did not exceed ca. 700 ppm. AB_5_ type hydrogen storage alloy powder (LaMmNi_4.1_Al_0.3_Mn_0.4_Co_0.45_) originated from VARTA Microbattery (Ellwangen, Germany) was obtained by ball-milling. Palladium(II) chloride (analytical grade, Alfa Aesar, London, UK) and ethylene glycol (analytical grade, Avantor Performance Materials, Gliwice, Poland) were used during Pd/AB_5_ composite synthesis (ca. 3.5 wt % of Pd in composite). 

### 4.2. Methods

AB_5_ alloy was decorated with Pd nanoparticles with the use of the modified microwave assisted polyol method described in detail in [11]. The electrochemical measurements were performed in a three-electrode system: LVE-working electrode (AB_5_, AB_5_-Pd-NPs), Ag wire—pseudoreference electrode, and Au plate as a counter electrode. The LVE electrode was prepared by pressing the alloy powder without binding additives into two Au meshes (6 mm diameter) on a hydraulic press under 6 t pressure for 3 min. AB_5_ alloy and AB_5_-Pd-NPs composite were electrochemically tested (Solartron SI 1287 Electrochemical Interface) by the means of cyclic voltammetry and chronopotentiometry. Before the main measurements, the alloy/composite was activated through continuous CV cycling in the potential range −1.1–1.2 V at scan rate 2 mV s^−1^ for 20 cycles and 20 CP deep charge and discharge steps. The electrode was charged and discharged at constant current 0.015 A g^−1^ (0.05 C). The physicochemical characteristics of the electrodes were ascertained with the use of X-ray photoelectron spectroscopy (XPS), scanning electron microscopy (SEM), and energy dispersive spectroscopy (EDS). A Microlab 350 (Thermo Electron, Waltham, MA, USA) equipped with a FEG-tip (Field-Emission Electron Gun) and a twin anode source (AlK_α_ and MgK_α_) was used for XPS analysis. XPS experiments were executed by MgKα (hv = 1253.6 eV) anode X-ray source operated at 15 kV and emission current intensity of 20 mA. The survey spectra and high-resolution spectra from the total surface area of 0.2 cm^2^ were recorded using 100 eV and 40 eV pass energy. Merlin scanning electron microscope (SEM, Zeiss, Germany) was used to examine the surface morphology of electrodes (EHT: 3 kV; probe current: 30 pA; system vacuum: 1.8 · 10^−9^ bar, gun vacuum: 9.5 · 10^−13^ bar). Energy dispersive X-ray spectroscopy (EDS; Quantax 400, Bruker, USA) was utilized to examine the composition of the electrodes (EHT 15 kV, probe current 500 pA). X-ray excitation with electron beam energy of 10–15 keV and spectrum acquisition time of ca. 100–120 s were applied. For energy calibration, X-ray signals obtained from pure Cu were utilized. Physicochemical characteristics of the ionic liquids, i.e., determination of the water content, density, dynamic viscosity, and ionic conductivity was done with the use of the following equipment: coulometric KF titrator without diaphragm (C10S, Mettler Toledo, Greifensee, Switzerland), densimeter (Densito, Mettler Toledo, Switzerland), rotary viscometer (ViscoQC300 with Peltier temperature device PTD 80 and CC12 measuring system, Anton Paar, Austria; rotational speed: 50 rpm), and conductometer (CX-401, Elmetron, Poland).

## Figures and Tables

**Figure 1 molecules-28-00856-f001:**
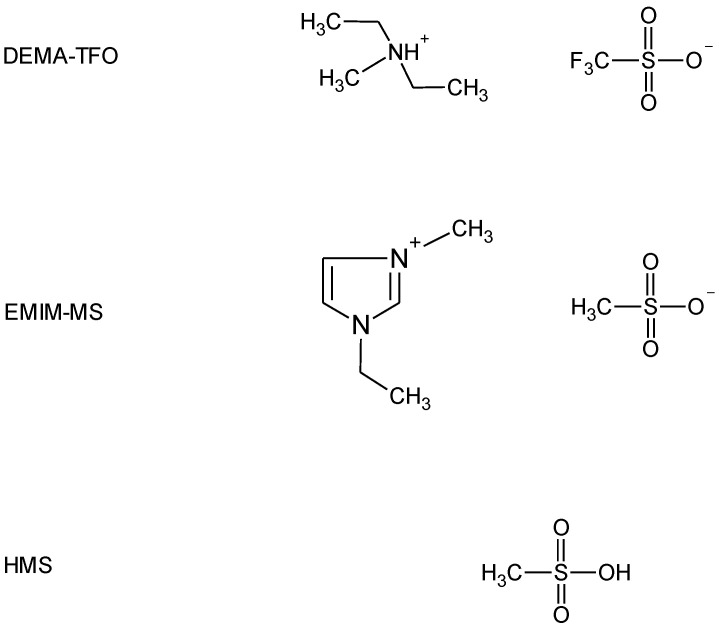
Structures of diethylmethylammonium triflate (DEMA-TFO), 1-ethyl-3-methylimidazolium methanesulfonate (EMIm-MS), and methanesulfonic acid (HMS) used in this study.

**Figure 2 molecules-28-00856-f002:**
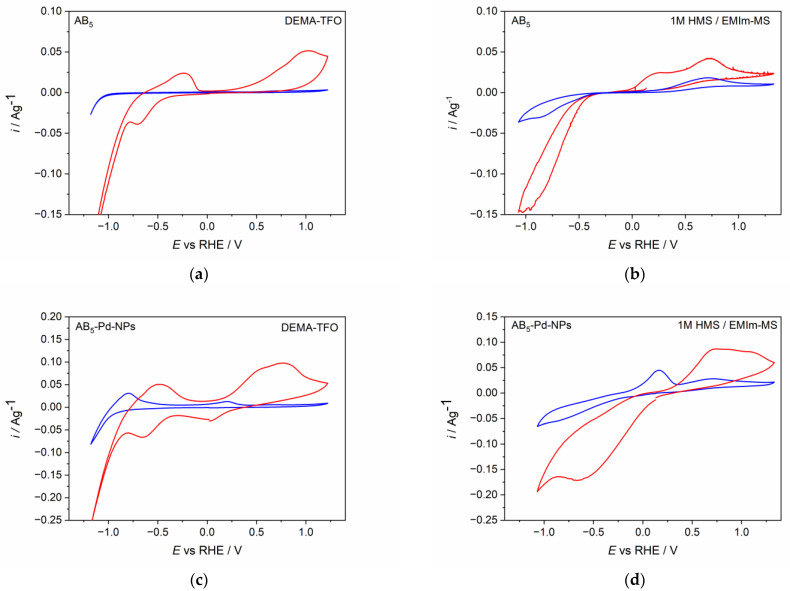
Cyclic voltammetry behavior of (**a**) AB_5_ in DEMA-TFO (**b**) AB_5_ in 1 M HMS/EMIm-MS (**c**) AB_5_-Pd-NPs in DEMA-TFO (**d**) AB_5_-Pd-NPs in 1 M HMS/EMIm-MS. Blue line—first CV cycle; red line—after activation (ca. 50 CV/CP cycles).

**Figure 3 molecules-28-00856-f003:**
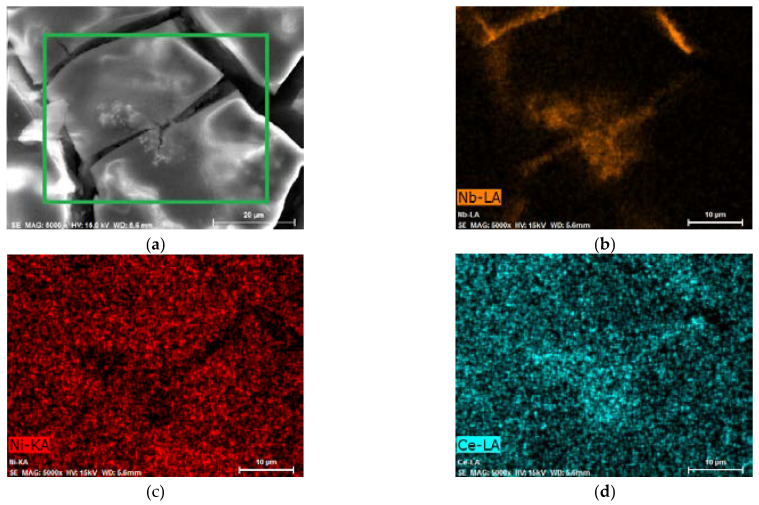
AB_5_ alloy electrode after electrochemical treatment in 1 M HMS/EMIm-MS (**a**) SEM image—EDS mapping for the area marked with a green rectangle; EDS map of (**b**) Nb—orange, (**c**) Ni—red, and (**d**) Ce—cyan. EDS mapping of other elements is presented in Appendix A.

**Figure 4 molecules-28-00856-f004:**
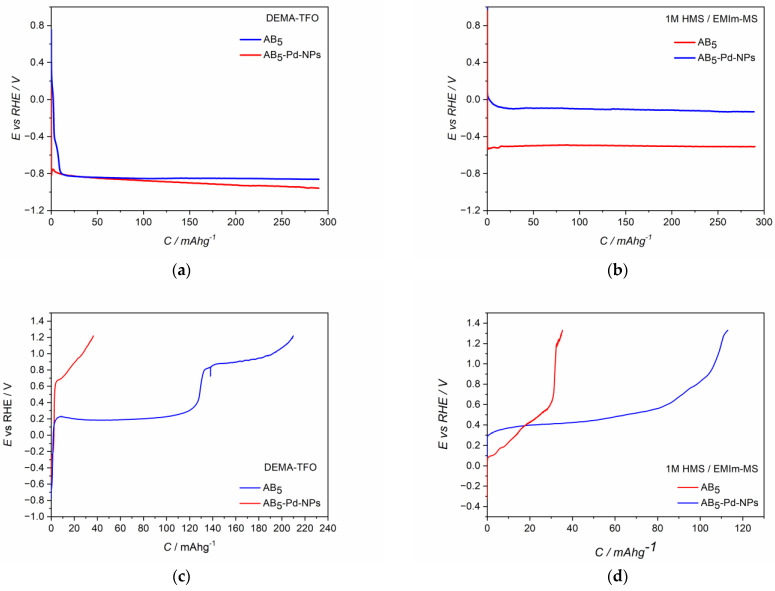
Charge (**a**,**b**) and discharge (**c**,**d**) potential profiles of electrodes in (**a**,**c**) DEMA-TFO; (**b**,**d**) 1 M HMS/EMIm-MS; Blue line—AB_5_-Pd-NPs; red line—AB_5_.

**Figure 5 molecules-28-00856-f005:**
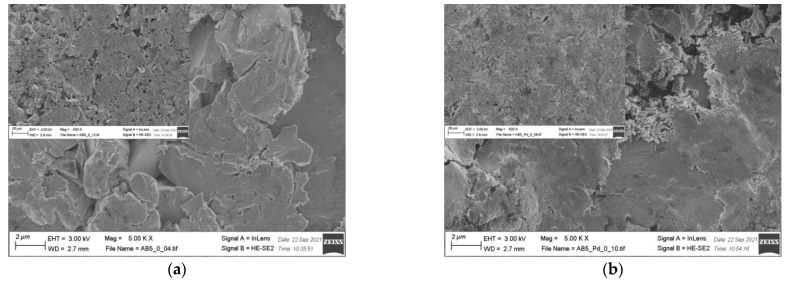
SEM images of (**a**) AB_5_; (**b**) AB_5_-Pd-NPs; (**c**) AB_5_ after electrochemical treatment in 1 M HMS/EMIm-MS; (**d**) AB_5_-Pd-NPs after electrochemical treatment in 1 M HMS/EMIm-MS; (**e**) AB_5_ after electrochemical treatment in DEMA-TFO; (**f**) AB_5_-Pd-NPs after electrochemical treatment in DEMA-TFO; the insets—magnification 500×.

**Table 1 molecules-28-00856-t001:** Physicochemical properties of ILs and its mixtures with parent superacid. For the electrochemical window, the values in brackets are: (cathodic limit /anodic limit).

IL	η/cP	ρ/gcm^−3^	σ/mScm^−1^	Water Content/wt. %, KF	Electrochemical Window/V
**DEMA-TFO**	58.97 ± 0.62	1.27 ± 0.01	7.69 ± 0.02	0.038 ± 0.001	(−1.1/3.0) 4.1
**1 M HMS/EMIm-MS**	181.70 ± 3.5	1.26 ± 0.01	1.66 ± 0.02	0.064 ± 0.002	(−1.9/2.0) 3.9

**Table 2 molecules-28-00856-t002:** Atomic concentrations (EDS) for AB_5_ and AB_5_-Pd-NPs electrodes before and after electrochemical treatment in DEMA-TFO and 1 M HMS/EMIm-MS.

Sample	EDS/% at.					
	Ni	La	Mn	Co	Al	Ce
pristine AB_5_	65.5 ± 4.0	9.4 ± 0.5	6.6 ± 0.4	8.3 ± 0.5	5.3 ± 0.3	4.9 ± 0.3
AB_5_-Pd-NPs	74.7 ± 4.0	8.2 ± 0.5	5.5 ± 0.3	7.8 ± 0.4	3.6 ± 0.4	4.2 ± 0.2
AB_5_ DEMA-TFO	63.3 ± 4.0	9.7 ± 0.5	8.0 ± 0.4	8.8 ± 0.5	4.2 ± 0.2	6.0 ± 0.3
AB_5_-Pd-NPs DEMA-TFO	71.8 ± 4.5	6.6 ± 0.3	4.4 ± 0.2	10.3 ± 0.6	3.3 ± 0.2	3.6 ± 0.2
AB_5_ 1 M HMS/EMIm-MS	56.7 ± 3.4	13.5 ± 0.7	9.0 ± 0.5	7.9 ± 0.4	4.4 ± 0.3	8.7 ± 0.4
AB_5_-Pd-NPs 1 M HMS/EMIm-MS	66.5 ± 4.0	7.9 ± 0.4	8.3 ± 0.4	9.2 ± 0.5	3.9 ± 0.2	4.2 ± 0.2

**Table 3 molecules-28-00856-t003:** Hydrogen absorption capacities calculated from chronopotentiometric discharging.

Sample	Capacity/mAh g^−1^	
	DEMA-TFO	1 M HMS/EMIm-MS
AB_5_	40 ± 5	40 ± 5
AB_5_-Pd-NPs	200 ± 10	110 ± 7

**Table 4 molecules-28-00856-t004:** Surface atomic concentrations (XPS) of selected elements for AB_5_ and AB_5_-Pd-NPs electrodes before and after electrochemical treatment in DEMA-TFO and 1 M HMS/EMIm-MS.

Sample	XPS/% at.					
	Ni	La	Mn	Co	Al	Ce
pristine AB_5_	47.5 ± 3.0	24.1 ± 1.5	10.1 ± 0.5	18.3 ± 1.0	-	-
AB_5_-Pd-NPs	70.1 ± 5.0	20.1 ± 1.5	-	9.8 ± 0.5	-	-
AB_5_ DEMA-TFO	43.0 ± 3.0	7.6 ± 0.4	7.6 ± 0.4	13.9 ± 0.7	27.8 ± 1.8	-
AB_5_-Pd-NPs DEMA-TFO	43.4 ± 3.0	9.4 ± 0.5	3.8 ± 0.2	9.4 ± 0.5	34.0 ± 2.0	-
AB_5_ 1 M HMS/EMIm-MS	50.3 ± 3.0	28.8 ± 2.0	10.8 ± 0.5	10.2 ± 0.5	-	-
AB_5_-Pd-NPs 1 M HMS/EMIm-MS	56.3 ± 3.5	8.3 ± 0.4	12.5 ± 0.7	8.3 ± 0.4	14.6 ± 0.9	-

## Data Availability

Not applicable.

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
