# Peer review of "The Electrochemical Behavior of Unmodified and Pd-NPs Modified AB5 Hydrogen Storage Alloy in Selected Protic and Aprotic Ionic Liquids (ILs): Towards ILs-Based Electrolytes for Ni-MH Batteries"

_molecules, 2023, doi:10.3390/molecules28020856_

Round 1

Reviewer 1 Report

Very nice, clear and well-written paper. I think it is worthy of being published with few modifications.

Abstract can be enlarged with more information about context, better highlighting the purpose of the work. 

In section 2, Authors stated that viscosity can affect proton transport, can the latter be quantified and proved some way?

Since it seems that wettability of the alloys may influence hydrogen electrosorption, future studies should be focused in better understanding this aspect. 

More information about stability of the employed electrolyte components should be provided. 

Some more details about parameters used for the adopted techniques that are only mentioned in section 4 should be provided (i.e., for SEM: EHT voltage, probe current, pressure, etc; for viscosimeter: shear rate or stress, etc, and so on).

Reviewer 2 Report

Review of paper molecules-2060241

Title: The electrochemical behaviour of unmodified and Pd-NPs modified AB5 hydrogen storage alloy in selected protic and aprotic ionic liquids (ILs): towards ILs-based electrolytes for Ni-MH batteries.

Comments to the authors:

The aim of this research article is to explore the use of protic or aprotic ionic liquids as electrolyte for Ni-MH based batteries. A protic ionic liquid (diethylmethylammonium triflate) and a mixture of an aprotic ionic liquid (1-ethyl-3-methylimidazolium methanesulfonate – DEMA-TFO) with parent superacid were used as electrolytes, with AB5 alloy electrodes and Pd nanoparticles-modified AB5 alloy electrodes.

Introduction part

The authors state that their previous study (https://doi.org/10.1016/j.elecom.2019.02.007, wrong DOI link within the pdf) shower better electrochemical properties toward hydrogen sorption of surface decorated AB5-type alloy with small amount of Pd nanoparticles. Therefore, they make only few comparisons of the performances within KOH and within the IL in the results part. However, if we look on the referred article, it is demonstrated that the kinetics of hydrogen desorption in the Pd NPs/AB5 composite were enhanced, and that activation was not required after modification of the AB5 alloy with Pd NPs. However, the electrochemical capacity is not studied in the related paper, the sentences related to this point are the following ones “In case of the pristine AB5 alloy LaMm-Ni4.1Al0.3Mn0.4Co0.45, the theoretical hydrogen sorption capacity does not exceed 290 mAh/g. This value was calculated from the hydrogen gas sorption experiment [26] and is the maximum value that could be obtained from electrochemical measurements in the case of pristine AB5. The addition of Pd to the composite does not significantly affect the capacity (there might in fact be a slight decrease).” Consequently, the sentence of the present article stating that these compounds have better electrochemical properties should be revised to be more precise. Moreover, it would improve the paper to have a real comparison between the behavior within the ionic liquids and within KOH. In particular, the capacity toward cycle number, not shown in KOH in this previous paper, should be added in this paper to be compared to the capacity toward cycle number in the different ionic liquids.

Part 2.1

In Table 1, it should be mentioned that it is weight % for the water content.

I would ask of the authors have measured the electrochemical windows of the two ionic liquids (pure or mixture with the parent acid)? If yes, is it possible to add this information within Table 1 and to compare them with the electrochemical window measured in the same conditions for KOH?

If not, it should be great that the authors make these measurements and complete the paper, as it is stated in the introduction part that the ionic liquids have wide electrochemical window.

It seems that the ionic liquids used in this study are commercial ones. Did the authors check the possible presence of impurities coming from the synthesis are an industrial scale?

Part 2.2

The curves of figure 2 are interesting. Could the author also compare them to the behavior within KOH (figure 2 in https://doi.org/10.1016/j.elecom.2019.02.007) ?

Page 4, line 108 : maybe from and not form (the signal from gaseous hydrogen…)?

Page 5, line 131 : worth and not wort (it is worth noting…)

I do not understand the sentence at the end of page 4. If the metal segregation process starts just after the synthesis, we should observe the related electrochemical signal in all the electrolytes? Could the author clarify this point please?

In the figure 3, it is necessary to better highlight the rectangular area of the mapping within the micrograph. The resolution of the EDS mapping is quite surprising: for the cerium L alpha map, we cannot see the cracks between the grains? For Ni K alpha map, we hardly see them. For Niobium L alpha map, it is correct. With this figure I can see the presence of Nb at some grain surfaces, but I cannot see the segregation of Ce at the grain surfaces? Do you think that the N in the central part of the map can be related to niobium oxides or hydroxides, the structures we see at the center of the micrograph? It could be important to map also the oxygen distribution map to understand this point. If I understand well, the EDS analysis has not been done on a polished surface? Why the other elements present within the alloy are not present in the EDS mapping?

I do not understand the fact that line 127 it is mentioned that Ce and Nb occur (not occurs) near the electrode surface, and line 132 and 133, La, Ni and Co segregate near the electrode surface whereas Ce and Al are in the depth of the grains?

The author notice that the electrolyte and the surface modification of the alloy may change (or not) the plateau potential of charging. But they do not discuss this point? Can a discussion be added?

Part 2.3

The table 2 should be gathered on one single page to help reading.

In this article, the effect of Pd nanoparticles seems to be real on the measured electrochemical capacity, compared to the paper https://doi.org/10.1016/j.elecom.2019.02.007. Could you discuss/explain why you have such difference? The measured capacities stay however low compared to the capacity of AB5 measured within KOH. This is shortly discussed line 176 page 7, the values within KOH should be given. Could you further discuss this difference and propose how you could improve the capacity within the ionic liquid electrolytes compared to KOH?

Do the authors measured the capacity for several cycles, to study the capacity evolution with cycle number?

Part 2.4

I do not understand on what type of samples the EDS measurements have been done? Is it polished cross section samples?

Table 3 and 4, it would be better if the authors could add uncertainty values for the different measurements. Why don’t we see the values for the Nb previously reported?

The authors state that “EDS results show that there is no significant change in the electrode composition before 200 and after Pd-NPs modification and after the electrochemical treatment in PIL/AIL electrolyte”. However, Ni content change from 65.5 to 63.3 and to 56.7. It seems to me it is quite a big change. Other elements quantification also changes. Could the author precise what they meant?

I do neither understand the XPS results: La stand from 24.1 in pristine to 7.6 at% in DEMA-TFO, it is not enriched. It grows up to 28.8 at% in AIL/acid, but the “bulk” La also increase from EDS measurement and it was stated it is stable. Could the author help to better understand what they wanted to explain?

Again, in the paragraph, enrichment in La, Co and Mn is stated, in contradiction with page 4 line 127 “more Ce and Nb occurs near the electrode surface”

The study of the surface with SEM micrographs after electrochemical cycling is interesting. How many electrochemical cycles before this analysis?

Figure 5.e highlights metal dissolution at the surface in DEMA-TFO. Do you think that the detection of Al at the surface after exposure to DEMA-TFO can be related to this dissolution?

In Figure 5, the scale for the insets can not be red. It is necessary to grow them up.

Reviewer 3 Report

The work is undoubtedly original: the electrochemical properties of the modified and unmodified AB5 hydrogen storage alloy in two ionic liquids are investigated.

The results are thoroughly discussed on the basis of the authors' previous works and supported by numerous references to the literature.

The work is correct in terms of content and I recommend its publication after minor corrections.

1. The most significant corrosion changes can be observed for the AB5 electrode after hydrogen electrosorption in DEMA-TFO (Fig. 5e). Such changes are not observed for the AB5 electrode after electrochemical hydrogenation processes in 1 M HMS/EMIm-MS (Fig. 5c). However, changes related to the greater surface degradation of the AB5 alloy particles in the DEMA-TFO electrolyte do not affect the discharge capacity of the AB5 electrode.

Please comment it.

2. The title of the journal is missing in the list of references (item 26).

Reviewer 4 Report

In this work, the authors studied the electrochemical behavior of unmodified and Pd-NPs modified AB5 hydrogen storage alloy within two superacid electrolytes of DEMA-TFO and 1 M HMS/EMIm-MS. However, the novelty is not very high, in addition, there are still some key issues not elucidated in the manuscript. I'm afraid I cannot recommend its publication in this journal. Some issues are listed as follows.

1.        Why 1 M HMS was used in the EMIm-MS system? Please give relevant explanation.

2.      The testing temperature of the physicochemical properties of the IL should be mentioned. The dynamic viscosity of DEMA-TFO in this study is 58.97 ± 0.62 cP, but the earlier study is 41.29 ± 0.08 cP (Journal of Alloys and Compounds 903 (2022) 163853). Please give specific explanation.

3.      Figure 2, the cycle number refers to the red and bule curves should be added, and quality of the figures should be improved.

4.        The discharge curves show two plateaus, where the lower one contributes to ~140 mAh g-1, and the higher plateau delivers ~80 mAh g-1. The corresponding capacity contribution from the Pd-NPs phase and the AB5 phase should be identified. In addition, the CV curves are suggested to correlate to the discussion.

5.        In the EDS and XPS study, the content of the modified Pd is missed.

6.        The XRD patterns are suggested to be supplemented to further confirm the surface corrosion.

Round 2

Reviewer 2 Report

I would like to thank the authors for their described responses and for the modification and correction of the article. The paper should be accepted for publication after some minor changes reported hereafter:

_ After the sentence "It is worth noting that the metal segregation process in AB5 particles starts just after the synthesis: La, Ni, Co tends to segregate near the electrode surface, whereas Ce and Al are present only in the depth of the grains.". Please add a sentence within the article such as precised in the response: 'it is well known phenomena for the AB-type alloys obtained by the ball-milling." Also, in the experimental section, please precise that the AB5 has been obtained by ball-milling, I think it is not precised anywhere?

I think that presenting all the maps and especially the O map is interesting, but I understand it takes too much space. A solution could be to add them in a supplementary material, do the authors think it could be done?

Author Response

We would like to thank the reviewer for the comments.

We have added the sentence about metal segregation to the article text. The information that AB5 has been obtained by ball-milling has been added to experimental section. All the EDS maps has been placed in supplementary material.

Reviewer 4 Report

The issues have been addressed, and the quality has been improved. The paper can be accpeted in current version.

Author Response

We would like to thank the reviewer for the comments, some minor corrections have been done according to the suggestions of other reviewer.